# Determinants of mammography screening participation–a cross-sectional analysis of the German population-based Gutenberg Health Study (GHS)

**Roman M. Pokora**[1‡]*, **Matthias Büttner**[1‡], **Andreas Schulz**[2], **Alexander K. Schuster**[3], **Hiltrud Merzenich**[1], **Andrea Teifke**[4], **Matthias Michal**[5,6], **Karl Lackner**[7], **Thomas Münzel**[8], **Sylke Ruth Zeissig**[9,10], **Philipp S. Wild**[2,6,11], **Susanne Singer**[1,12], **Daniel Wollschläger**[1]

1 Institute for Medical Biostatistics, Epidemiology and Informatics, University Medical Center of the Johannes Gutenberg University Mainz, Mainz, Germany, 2 Preventive Cardiology and Preventive Medicine, Department of Cardiology, University Medical Center of the Johannes Gutenberg University Mainz, Mainz, Germany, 3 Department of Ophthalmology, University Medical Center of the Johannes Gutenberg University Mainz, Mainz, Germany, 4 Klinik und Poliklinik für Diagnostische und Interventionelle Radiologie, University Medical Center of the Johannes Gutenberg University Mainz, Mainz, Germany, 5 Department of Psychosomatic Medicine and Psychotherapy, University Medical Center of the Johannes Gutenberg University Mainz, Mainz, Germany, 6 German Center for Cardiovascular Research (DZHK), partner site Rhine-Main, Mainz, Germany, 7 Institute for Clinical Chemistry and Laboratory Medicine, University Medical Center of the Johannes Gutenberg University Mainz, Mainz, Germany, 8 Department of Cardiology, Cardiology I, University Medical Center of the Johannes Gutenberg University Mainz, Mainz, Germany, 9 Institute of Clinical Epidemiology and Biometry, University of Würzburg, Würzburg, Germany, 10 Regional Centre Würzburg, Bavarian Cancer Registry, Bavarian Health and Food Safety Authority, Würzburg, Germany, 11 Center for Thrombosis and Hemostasis (CTH), University Medical Center of the Johannes Gutenberg University Mainz, Mainz, Germany, 12 University Cancer Center Mainz, Mainz, Germany

‡ These authors share first authorship on this work.
* pokora@uni-mainz.de

## Abstract

### Purpose

We investigated the association between social inequality and participation in a mammography screening program (MSP). Since the German government offers mammography screening free of charge, any effect of social inequality on participation should be due to educational status and not due to the financial burden.

### Methods

The 'Gutenberg Health Study' is a cohort study in the Rhine-Main-region, Germany. A health check-up was performed, and questions about medical history, health behavior, including secondary prevention such as use of mammography, and social status are included. Two indicators of social inequality (equivalence income and educational status), an interaction term of these two, and different covariables were used to explore an association in different logistic regression models.

### Results

A total of 4,681 women meeting the inclusion criteria were included. Only 6.2% never participated in the MSP. A higher income was associated with higher chances of ever participating

**Data Availability Statement:** The GHS data cannot be made publicly available because participants

were assured that the data would only be used for scientific research purposes, and therefore the informed consent included only limited data sharing. At GHS, there is a separate consent to share data with collaborators. However, if this consent is given, the data can only be shared as part of a scientific collaboration if the GHS Steering Committee deems it appropriate and necessary from a scientific perspective. However, there are also participants who have not consented to the sharing of data in principle. For these participants, data can only be analyzed on-site. Available data are available to researchers who meet the criteria for access to confidential data from the GHS. More detailed contact information can be found on the home pages of the GHS (http://www.gutenberghealthstudy.org/ghs/overview.html?L=1) or per mail info@ghs-mainz.de.

**Funding:** "The GHS is funded through contract AZ 961-386261/733 from the government of Rhineland-Palatinate ("Stiftung Rheinland-Pfalz für Innovation"); the research programs "Wissen schafft Zukunft" and "Center for Translational Vascular Biology" of the Johannes Gutenberg University Mainz; and its contract with Boehringer Ingelheim and Philips Medical Systems, including an unrestricted grant for the GHS. This study was also supported by grant BMBF 01EO1503 from the Federal Ministry of Education and Research. The funders had no role in study design, data collection and analysis, decision to publish, or preparation of the manuscript."

**Competing interests:** "Dr. Wild reports grants and personal fees from Boehringer Ingelheim, grants and personal fees from Novartis Pharma, grants from Philips Medical Systems, grants from Bayer AG, grants and personal fees from Sanofi-Aventis, grants and personal fees from Bayer Vital, grants and personal fees from Daiichy Sankyo, grants and personal fees from Bayer Health Care, personal fees from AstraZeneca, personal fees and non-financial support from DiaSorin, non-financial support from I.E.M., outside the submitted work; Dr. Schuster reports the professorship for ophthalmic healthcare research endowed by 'Stiftung Auge' and financed by 'Deutsche Ophthalmologische Gesellschaft' and 'Berufsverband der Augenarzte Deutschlands e.V.' Schuster AK received research funding from Allergan, Bayer Vital, Novartis, PlusOptix and Heidelberg Engineering; Dr. Singer reports personal fees from Pfizer, personal fees from Bristol-Myers Squibb, personal fees from Boehringer-Ingelheim, personal fees from Lilly, outside the submitted work; Dr. Wollschläger reports grants from German Federal Ministry of

in a mammography screening (odds ratios (OR): 1.67 per €1000; 95%CI:1.26–2.25, model 3, adjusted for age, education and an interaction term of income and education). Compared to women with a low educational status, the odds ratios for ever participating in the MSP was lower for the intermediate educational status group (OR = 0.64, 95%CI:0.45–0.91) and for the high educational status group (0.53, 95%CI:0.37–0.76). Results persisted also after controlling for relevant confounders.

## Conclusions

Despite the absence of financial barriers for participation in the MSP, socioeconomic inequalities still influence participation. It would be interesting to examine whether the educational effect is due to an informed decision.

## Introduction

At present, about 70,000 women in Germany are newly diagnosed with breast cancer each year, and 18,000 die due to this disease [1]. Hence, breast cancer is the most common cancer in women in Germany. Mammography is currently the most effective method of detecting breast cancer at a prognostically favorable stage, which is not yet possible during a palpation examination [2].

Since 2009, the Mammography Screening Program (MSP) has been offered nationwide in Germany for women aged 50 to 69 years, with the aim to reduce breast cancer mortality [3–7]. Eligible women are informed every two years by an invitation letter from one of 14 Central Offices which organize the mammography screening program nationwide. A detailed information brochure is sent out with the invitation. The brochure serves as a decision-making aid and provides information on the examination procedure, on breast cancer, and on possible advantages as well as disadvantages of participating in a screening program. The European Guidance schedule sets a 70% participation in the MSP as a quality target [8]. However, like any screening examination, mammography carries the risk of overdiagnosis. Other pros and cons of mammography are still discussed and there is a controversial discussion on the actual merit of regular mammographic examinations in general, and of a mammography screening program in particular [7]. In Germany, the Patients' Rights Act states that no medical intervention may be carried out without an informed decision. The Institute for Quality and Efficiency in Health Care provides decision support material to help women weigh their individual advantages and disadvantages of participating in the MSP. This decision support material is intended to illustrate the positive consequences of the MSP. However, participation in the MSP declined following the introduction of the new leaflet [9].

Studies on social inequality and health have shown that a lower socio-economic status (generally measured by combining levels of education, income and/or professional position) is associated with higher rates of morbidity and mortality. Factors that might explain these health inequalities are health-risk behaviors, monetary and psychosocial disadvantages, stressors, and deficiencies in health care [10, 11]. Regarding persons of higher age, there is little evidence of status-specific differences in health care supply compared to the younger population [12–14]. Knesebeck and Mielck [14] were able to show a higher association with participation in mammography in the last two years for both higher education and higher equivalence income as well as higher monetary wealth.

Education and Research, during the conduct of the
study. All other authors declare no conflict of
interest. This does not alter our adherence to PLOS
ONE policies on sharing data and materials."

Irregular participation in the MSP is a public health concern [15, 16]. Research has shown that there are regional, demographic, socio-economic, and educational as well as behavioral differences in adherence to organized mammography screening [17–19]. In addition, there was a clear urban-rural gradient across Europe, with lower participation in urban than in rural areas [16, 20]. In a German study on attendance in the MSP, 20,000 women in Northern Germany were contacted [21]. Among other questions, they were asked about reasons for non-participation, with more than 40% of respondents mentioning medical reasons or personal attitude. Concrete reasons ranged from wanting to be further examined by the previous physician, already taking part in an annual mammography examination elsewhere (examples of medical reasons), to having private insurance, and feeling that mammography is too painful (examples of personal attitude). The timing of the MSP examination and a lack of information were rarely mentioned.

Social inequality regarding participation in screening program among people eligible for screening is seldom explored, and women's attitudes towards mammography have changed over time. Therefore, we expect an association between socioeconomic status and mammography similar to that found in the study by Knesebeck and Mielck [14]. Furthermore, we investigated the association between socioeconomic differences and MSP participation and controlled for covariables which also had an impact on participation in mammography.

## Materials and methods

### Study sample

The Gutenberg Health Study (GHS) is a population-based, prospective, observational, single center cohort study in the Rhein Main Region in western Germany that includes 15,010 participants from the general population. The study sample was randomly drawn from local registry offices with a participation rate of 60% [22]. The GHS was approved by the ethics committee of the Medical Chamber of Rhineland-Palatine, Germany, and was conducted in accordance with the declaration of Helsinki. All participants provided written informed consent before entering the study.

The focus of the GHS Study is on evaluating cardiovascular risk factors and estimating the incidence of cardiovascular diseases in the general population. Study participants pass through a standardized cardiovascular examination program at University Medical Center, Mainz. In addition to the physical examinations, participants complete a computer-assisted personal interview to assess sociodemographic variables, prescribed medications, medical conditions diagnosed by a physician, lifestyle-factors such as smoking, and family history of cardiovascular and malignant diseases. Furthermore, subjects completed questionnaires on physical activity, environmental and occupational factors, quality of life, and mental health.

This analysis was performed with a sub-sample of the GHS. Participants were included in our analysis if they were female, provided information on their mammography screening status, and were between 50 and 69 years of age at their baseline examination. In that age group, women in Germany are actively invited to participate in the MSP and the costs are fully covered by statutory health insurance. Since the GHS standard examinations took place between 2007 and 2012, every woman in her 50s had been invited to the MSP at least once.

### Variables for the analysis

Information on participation in a mammography screening was obtained by asking the participant if a mammography screening for cancer prevention was ever performed.

Education of the participant was defined as the highest obtained school degree according to the German school system. For the assessment of income, participants were asked to state

their total monthly net household income, which was then adjusted according to the OECD equivalence scale [23], resulting in four categories: <€1000, €1000 to <€1500, €1500 to <€3000, and ≥€3000. Participants were categorized as either living in a rural or an urban setting. Living with a partner or living alone were the distinctions in the partnership variable. Participants were insured by statutory health insurance or private health insurance, with the statutory health insurance representing the main type of health insurance in Germany. Migration status was defined in accordance with the German census, resulting in three categories: non-migrants, first generation migrants, and second generation migrants [24]. First generation migrants are all who migrated to Germany after 1949, while second generation migrants are all non-German citizens born in Germany and all citizens born in Germany with at least one parent migrated or living abroad.

Smoking status was determined with the help of a structured interview about smoking behavior. Alcohol consumption was defined as being above the limit or below with the limit being defined by the WHO definition (≥10g/day for women) [25].

Working status was assessed by asking if the participant was currently working (full and part-time).

In addition, diagnosis of breast cancer or any other cancer of the patient and a breast cancer diagnosis for the mother below 50 years of age was obtained using free text questions.

## Statistical analysis

Sociodemographic and medical characteristics of the participants were expressed by mean values for continuous variables and by relative and absolute frequencies for discrete variables.

Multiple logistic regression was performed with "having ever taken part in a mammography screening" as the outcome variable. Five models were created to evaluate the association between mammography screening and sociodemographic and medical factors. The models included the following covariates, respectively: (1) age and education; (2) age and income; (3) age, education, income, and an interaction term for education and income; (4) age, education, income, an interaction term for education and income, breast cancer diagnosis of the mother, health insurance, number of children, breast feeding, BMI, smoking status, alcohol consumption, partnership status, living residence, working status, and migration status; (5) age, education, income, an interaction term for education and income, breast cancer diagnosis of the mother, health insurance, number of children, breast feeding, BMI, smoking status, alcohol consumption, partnership status, living residence, working status, migration status; breast cancer diagnosis, and other cancer diagnosis. Covariates were selected based on the literature and the available data within the study sample. Goodness-of-fit of the models was assessed using the likelihood-ratio-test. Missing values in the prediction variables were imputed using the missForest R package. All statistical analysis was performed using R version 3.6.0 (R Foundation for Statistical Computing, Vienna, Austria).

## Results

In total, 7426 participants of the GHS sample were female (mean age:54.8, standard deviation (SD):11.1). Twenty-seven (0.4%) participants had missing information on mammography screening or answered, "don't know". 2718 (36.6%) were excluded from the analysis because they did not fit the age range where mammography screening is recommended, resulting in 4681 women eligible for this analysis. Of those 4681 participants, 290 (6.2%) reported not having had any mammography screening. The mean age of the study population was 61.4 (SD:7.0) years. All characteristics for the entire study population, participants without mammography and participants with mammography are presented in Table 1.

**Table 1. Sample characteristics.**

| Characteristic<br>mean (SD) or n (%) or mean [95%CI] | All participants (n = 4681) | Mammography screening (n = 4391) | No mammography screening (n = 290) |
|---|---|---|---|
| **Age (years)** | 61.4 (7.0) | 61.4 (7.0) | 61.3 (7.7) |
| **Living setting** | | | |
| Urban | 2211 (47.2%) | 2097 (47.8%) | 114 (39.3%) |
| Rural | 2470 (52.8%) | 2294 (52.2%) | 176 (60.7%) |
| **Education** | | | |
| <10 years of schooling | 2355 (50.5%) | 2227 (50.9%) | 128 (44.4%) |
| 10 years of schooling | 1192 (25.6%) | 1115 (25.5%) | 77 (26.7%) |
| > 10 years of schooling | 1040 (22.3%) | 966 (22.1%) | 74 (25.7%) |
| Other education | 36 (0.8%) | 31 (0.7%) | 5 (1.7%) |
| No graduation | 36 (0.8%) | 32 (0.7%) | 4 (1.4%) |
| Missing | 22 | 20 | 2 |
| **Equivalence income** | | | |
| <€1000 | 531 (12.8%) | 478 (12.3%) | 53 (20.9%) |
| €1000 to €1499 | 745 (17.9%) | 686 (17.6%) | 59 (23.2%) |
| €1500 to €2999 | 2125 (51.2%) | 2021 (51.9%) | 104 (40.9%) |
| ≥€3000 | 750 (18.1%) | 712 (18.3%) | 38 (15.0%) |
| Missing | 530 | 494 | 36 |
| **Migration** | | | |
| No migration | 3603 (77.0%) | 3400 (77,5%) | 203 (70.0%) |
| 1th generation migrant | 468 (10.0%) | 423 (9.6%) | 45 (15.5%) |
| 2th generation migrant | 608 (13.0%) | 566 (12.9%) | 42 (14.5%) |
| Missing | 2 | 2 | 0 |
| **Private health insurance** | | | |
| Yes | 754 (16.1%) | 711 (16.2%) | 43 (14.8%) |
| No | 3923 (83.9%) | 3676 (83.8%) | 247 (85.2%) |
| Missing | 4 | 4 | 0 |
| **Living in partnership** | | | |
| Yes | 3606 (77.1%) | 3416 (77.8%) | 190 (65.5%) |
| No | 1074 (22.9%) | 974 (22.2%) | 100 (34.5%) |
| Missing | 1 | 1 | 0 |
| **Children** | | | |
| Number of children | 1.9 (1.0) | 1.9 (0.9) | 2.1 (1.1) |
| Missing | 590 | | |
| **Breast feeding** | | | |
| Yes | 2601 (63.9%) | 2437 (63.8%) | 164 (65.6%) |
| No | 1467 (36.1%) | 1381 (36.2%) | 86 (34.4%) |
| Missing | 613 | 573 | 40 |
| **Body-Mass-Index (kg/m$^2$)** | 26.6 [23.5;30.7] | 26.6 [23.5;30.7] | 26.9 [23.2;31.2] |
| **Current smoker** | | | |
| Yes | 699 (14.9%) | 638 (14.6%) | 61 (21.0%) |
| No | 3973 (85.1%) | 3744 (85.4%) | 229 (79.0%) |
| Missing | 9 | 9 | 0 |
| **Alcohol consumption (above the limit)** | | | |
| Yes | 1177 (25.2%) | 1130 (25.8%) | 47 (16.2%) |
| No | 3494 (74.8%) | 3251 (74.2%) | 243 (85.8%) |
| Missing | 10 | 10 | 0 |

*(Continued)*

**Table 1.** (Continued)

| Characteristic | All participants (n = 4681) | Mammography screening (n = 4391) | No mammography screening (n = 290) |
|---|---|---|---|
| **mean (SD) or n (%) or mean [95%CI]** | | | |
| **Currently working** | | | |
| Yes | 1685 (36.2%) | 1576 (36.1%) | 109 (37.8%) |
| No | 2971 (63.8%) | 2792 (63.9%) | 179 (62.2%) |
| Missing | 25 | 23 | 2 |
| **Diagnosed with cancer** | | | |
| Yes | 611 (13.1%) | 595 (13.6%) | 16 (5.5%) |
| No | 4064 (86.9%) | 3791 (86.4%) | 273 (94.5%) |
| Missing | 6 | 5 | 1 |
| **Diagnosed with breast cancer** | | | |
| Yes | 237 (5.1%) | 236 (5.4%) | 1 (0.3%) |
| No | 4438 (94.9%) | 4150 (94.6%) | 288 (99.7%) |
| Missing | 6 | 5 | 1 |
| **Mother diagnosed with breast cancer below age 50** | | | |
| Yes | 47 (1.0%) | 46 (1.0%) | 1 (0.3%) |
| No | 4644 (99.0%) | 4345 (99%) | 289 (99.7%) |

% were calculated excluding missing values

Table 2 shows the results of multiple logistic regression analyses for participating in a mammography screening for breast cancer prevention. Across all models, increasing age was not significantly associated with the chance of having ever participated in a mammography screening. Participants with middle or high education were significantly less likely to participate in a mammography screening compared to participants with low education level. Across all models, a higher income was associated with higher odds (odds ratio (OR): 1.25–1.67 (per €1000); $p < 0.005$ for all models) for attending a mammography screening. Women living in a partnership (compared to no partnership), living in an urban setting (compared to a rural setting), reporting alcohol consumption above the limit (compared to below the limit) were significantly more likely to participate in a mammography screening, while those who smoke (compared to non-smokers) were less likely to participate in the screening (models 4 and 5). First generation migrants were less likely to have participated in a mammography screening compared to non-migrants, while there was no significant effect for second generation migrants compared to non-migrants. Participants who had a breast cancer diagnosis were more likely to participate in mammography screening, as well as participants with other cancer diagnosis (model 5). No significant differences in screening participation were seen for health insurance status, breast-feeding, BMI, working status, and the interaction of education and income. The likelihood-ratio-test indicated the best fit for the model with the highest number of covariates (model 5 vs. model 1,3,4 $P < 0.0001$; model 2 is not part of the other models, so it cannot be compared).

## Discussion

The aim of this study was to discover determinants of participation in the German MSP. Therefore, the participation in a mammography screening within the last two years for the eligible age group was assessed. Only 6.2% of our study participant reported having never participated in a mammography screening.

**Table 2. Multiple logistic regression models for having taken part in a mammography screening for breast cancer (dependent variable).**

| Independent variables | Model 1 | Model 2 | Model 3 | Model 4 | Model 5 |
|---|---|---|---|---|---|
| **Age (10 year increase)** | 0.99 [0.83;1.18] | 1.09 [0.92;1.30] | 1.05 [0.88;1.25] | 1.01 [0.81;1.27] | 0.98 [0.78;1.23] |
| **Education (Reference <10 years of schooling)** | | | | | |
| No graduation | 0.46 [0.18;1.56] | | 0.79 [0.12;32.42] | 1.11 [0.18;43.55] | 0.97 [0.17;35.61] |
| Ten years of schooling | 0.82 [0.62;1.11] | | 0.64 [0.45;0.90] | 0.64 [0.45;0.91] | 0.64 [0.45;0.91] |
| More than ten years of schooling | 0.75 [0.55;1.02] | | 0.53 [0.37;0.76] | 0.55 [0.37;0.82] | 0.54 [0.36;0.81] |
| Other education | 0.36 [0.15;1.06] | | 0.39 [0.11;2.48] | 0.37 [0.10;2.58] | 0.33 [0.09;2.42] |
| **Equivalence income (€1000 centered)** | | 1.25 [1.11;1.42] | 1.67 [1.26;2.25] | 1.42 [1.08;1.93] | 1.43 [1.09;1.94] |
| **Income (€1000 centered)* Education (Reference <10 years of schooling)** | | | | | |
| Income* no graduation | | | 1.31 [0.25;22.05] | 1.31 [0.27;21.78] | 1.22 [0.28;19.01] |
| income*ten years of schooling | | | 0.79 [0.54;1.16] | 0.80 [0.56;1.16] | 0.81 [0.56;1.17] |
| income*more than ten years of schooling | | | 0.72 [0.51;1.01] | 0.77 [0.55;1.06] | 0.77 [0.55;1.08] |
| income*other education | | | 1.01 [0.25;5.18] | 0.95 [0.22;5.40] | 0.91 [0.20;5.55] |
| **Breast cancer mother below age 50 (vs. no breast cancer)** | | | | 3.43 [0.73;61.22] | 3.39 [0.72;60.43] |
| **Private insurance (vs. statutory insurance)** | | | | 0.97 [0.67;1.45] | 0.97 [0.66;1.44] |
| **No children (vs. 1–2 children)** | | | | 0.79 [0.41;1.69] | 0.81 [0.42;1.73] |
| **>2 children (vs. 1–2 children)** | | | | 0.64 [0.48;0.85] | 0.67 [0.50;0.89] |
| **Breast feeding (vs. no breast feeding)** | | | | 0.99 [0.74;1.30] | 0.98 [0.74;1.30] |
| **BMI (kg/m2)** | | | | 1.01 [0.98;1.03] | 1.01 [0.99;1.03] |
| **Smoking (vs. no smoking)** | | | | 0.61 [0.45;0.84] | 0.61 [0.45;0.84] |
| **Alcohol above limit (vs. below limit)** | | | | 1.71 [1.24;2.40] | 1.73 [1.25;2.43] |
| **In partnership (vs. no partnership)** | | | | 1.66 [1.26;2.18] | 1.68 [1.27;2.20] |
| **Urban residence (vs. rural residence)** | | | | 1.60 [1.25;2.07] | 1.60 [1.24;2.07] |
| **Working (vs. not working)** | | | | 0.99 [0.72;1.38] | 1.01 [0.73;1.41] |
| **1th generation migrant (vs. non migrant)** | | | | 0.65 [0.46;0.96] | 0.67 [0.47;0.98] |
| **2th generation migrant (vs. non migrant)** | | | | 0.82 [0.58;1.19] | 0.80 [0.57;1.16] |
| **Cancer self (vs. no cancer)** | | | | | 1.95 [1.16 3.55] |
| **Breast cancer (vs. no breast cancer)** | | | | | 8.22 [1.63;149.83] |

All results expressed as OR [95%CI]; Reference variables in ()

The study shows that women with a higher income had higher mammography attendance in Mainz and the surrounding area. Women with higher education had lower attendance. These effects persist after the introduction of an interaction term of both variables and are even further reinforced by this. Based on the available results, it is possible to confirm the frequent tendency that people with a comparatively high educational status participate less often in mammography than people with a low educational status [14, 26, 27]. In the work of Knesebeck and Mielk [14], a positive correlation with income and participation in mammography screening was shown. For the income we found a different relationship. This indicates that the reasons for participation or non-participation may have changed over time. It seems that people with increasing education have reasons not to participate in a mammography. It became apparent, especially after the introduction of the new decision support material, that the number of participants decreased [9]. The decline may represent an informed decision in the higher educated groups, possibly reflecting, for example, fear of radiation exposure.

Compared to other studies, we were able to adjust for various factors which possibly affect participation in a mammography, such as whether the mother ever had breast cancer. Depending on the genetic mutation, women with a genetic predisposition face a lifetime risk for breast cancer of up to 65 to 80% [28]. These women should therefore have a mammography on a

regular basis. As expected, there was a strong association between participation and women whose mother was diagnosed with breast cancer and participation in a mammography.

Other factors, which are protective for breast cancer, such as the number of children, breastfeeding [29], or body weight [30] had no association with participation rates. Smoking and drinking above the limit defined to be within healthy behavior both had an association with participation rates. While the association of smoking with participation in mammography is rather negative, there is a positive connection for alcohol over the limit. The positive connection between high alcohol consumption and participation in a mammography could be due to the fact that a rather large number of women in the wine region of Mainz, especially among the elderly, consumed alcohol in an amount higher than the tolerable upper alcohol intake level, which may be harmful to health [31].

Not being married, not being German, and living in the countryside was also associated with a higher chance of not having participated in a mammography. The positive association between the first generation of a migrant background on the one hand and the place of residence in rural or urban areas on the other hand with participation in a mammography may show some sort of access restriction. In quantitative studies with migrants, language and transportation problems are the most commonly perceived barriers. First-generation migrants often must rely on their husband or their children for reading the invitation letter [32]. The reasons for the differences in the participation between women living in a city and women living in the country is for Germany still unclear. In a survey of 20,000 women in northern Germany who did not participate in mammography screening after an invitation letter, no women mentioned transportation problems [21].

The main limitation of our study pertains to the cross-sectional data acquisition and the fact that only German speaking migrants were able to participate. The results are based on survey data. Information on participation in a mammography was self-reported, which implies a risk of response distortions and incorrect answers (e.g. due to memory errors or incorrect information on income).

The strengths of the study are the well characterized population of participants between 50 and 74 years living in the Rhine-Main Region in Germany and the relatively large sample size. The GHS overall had a reasonable response rate of 60.3%. The study is therefore in the usual framework for population surveys. For example, in the German part of the 'Survey of Health, Ageing and Retirement in Europe', the response rate was 60.2% [14]. In this study, the non-responders may introduce bias in our study, as they tended to be slightly older and maybe less willing to participate also in a mammography. This could, at least in part, be attenuated by our study center's easy accessibility via free public transport. In addition, concerns about mammography shouldn't play a role in our study. Unlike previous studies [14], we were also able to adjust statistically for various factors.

In conclusion, socioeconomic inequalities may still influence participation in the German mass-screening program. The reasons for this are not clear, and reasons for the participation are still changing. For example, in 2015 a new leaflet was introduced, and participation rates declined afterwards [9]. The results emphasize the need to repeat analyses of the social differences in health care on a regular timed basis to evaluate public health measures.

## Acknowledgments

We thank all participants in the GHS and the study team, including study assistants, interviewers, computer and laboratory technicians, research scientists, managers, and statisticians.

## Author Contributions

**Conceptualization:** Roman M. Pokora, Matthias Büttner, Hiltrud Merzenich, Andrea Teifke, Sylke Ruth Zeissig, Susanne Singer, Daniel Wollschläger.

**Data curation:** Roman M. Pokora, Matthias Büttner, Andreas Schulz, Alexander K. Schuster.

**Formal analysis:** Roman M. Pokora, Matthias Büttner, Andreas Schulz, Alexander K. Schuster, Susanne Singer, Daniel Wollschläger.

**Funding acquisition:** Matthias Michal, Karl Lackner, Thomas Münzel, Philipp S. Wild.

**Investigation:** Roman M. Pokora.

**Methodology:** Roman M. Pokora, Matthias Büttner, Susanne Singer, Daniel Wollschläger.

**Project administration:** Roman M. Pokora, Matthias Michal, Karl Lackner, Thomas Münzel, Philipp S. Wild.

**Resources:** Philipp S. Wild.

**Software:** Andreas Schulz.

**Supervision:** Alexander K. Schuster, Sylke Ruth Zeissig, Philipp S. Wild, Susanne Singer, Daniel Wollschläger.

**Validation:** Andreas Schulz.

**Visualization:** Andreas Schulz.

**Writing – original draft:** Roman M. Pokora, Matthias Büttner.

**Writing – review & editing:** Andreas Schulz, Alexander K. Schuster, Hiltrud Merzenich, Andrea Teifke, Matthias Michal, Karl Lackner, Thomas Münzel, Sylke Ruth Zeissig, Philipp S. Wild, Susanne Singer, Daniel Wollschläger.

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
