## [Decision Letter · Decision Letter 0]

17 Feb 2022

PONE-D-21-26679Determinants of mammography screening participation – A cross-sectional analysis of the German population-based Gutenberg Health Study (GHS)PLOS ONE

Dear Dr. Roman Pokora,

Thank you for submitting your manuscript to PLOS ONE. After careful consideration, we feel that it has merit but does not fully meet PLOS ONE’s publication criteria as it currently stands. Therefore, we invite you to submit a revised version of the manuscript that addresses the points raised during the review process.

Thank you for your interesting article, but it needs Minor Revision, as suggested by expert reviewers. Please address the concerns raised by the reviewers and resubmit the revised version. 

We look forward to receiving your revised manuscript.

Kind regards,

Sajid Bashir Soofi

Academic Editor

PLOS ONE

https://journals.plos.org/plosone/s/fileid=ba62/PLOSOne_formatting_sample_title_authors_affiliations.pdf".

“The GHS is funded through contract AZ 961-386261/733 from the government of Rhineland-Palatinate (“Stiftung Rheinland-Pfalz für Innovation”); the research programs “Wissen schafft Zukunft” and “Center for Translational Vascular Biology” of the Johannes Gutenberg University Mainz; and its contract with Boehringer Ingelheim and Philips Medical Systems, including an unrestricted grant for the GHS. This study was also supported by grant BMBF 01EO1503 from the Federal Ministry of Education and Research.”

“Dr. Wild reports grants and personal fees from Boehringer Ingelheim, grants and personal fees from Novartis Pharma, grants from Philips Medical Systems, grants from Bayer AG, grants and personal fees from Sanofi-Aventis, grants and personal fees from Bayer Vital, grants and personal fees from Daiichy Sankyo, grants and personal fees from Bayer Health Care, personal fees from AstraZeneca, personal fees and non-financial support from DiaSorin, non-financial support from I.E.M., outside the submitted work; Dr. Schuster reports the professorship for ophthalmic healthcare research endowed by ‘Stiftung Auge’ and financed by ‘Deutsche Ophthalmologische Gesellschaft’ and ‘Berufsverband der Augenarzte Deutschlands e.V.’ Schuster AK received research funding from Allergan, Bayer Vital, Novartis, PlusOptix and Heidelberg Engineering; Dr. Singer reports personal fees from Pfizer, personal fees from Bristol-Myers Squibb, personal fees from Boehringer-Ingelheim, personal fees from Lilly, outside the submitted work; Dr. Wollschläger reports grants from German Federal Ministry of Education and Research, during the conduct of the study. All other authors declare no conflict of interest.”

Additional Editor Comments:

Please address the point-by-point comments raised by reviewers.

Reviewers' comments:

Reviewer's Responses to Questions

**Comments to the Author**

1. Is the manuscript technically sound, and do the data support the conclusions?

Reviewer #1: Yes

Reviewer #2: Yes

2. Has the statistical analysis been performed appropriately and rigorously? 

Reviewer #1: I Don't Know

Reviewer #2: I Don't Know

3. Have the authors made all data underlying the findings in their manuscript fully available?

Reviewer #1: Yes

Reviewer #2: No

4. Is the manuscript presented in an intelligible fashion and written in standard English?

Reviewer #1: Yes

Reviewer #2: Yes

5. Review Comments to the Author

Reviewer #1: It is a technically sound study, well conducted and a well written manuscript. Would be nice to add a little detail about the mammography invitation process for the non-German reader. I feel it would be important to explore perhaps as a separate study, the reasons behind the decrease in mammography (recently) with higher education. Are they accessing bad information on social media or misinterpreting information about radiation exposure etc. This may inform further decisions and the invitation process may need to have more detailed information on dispelling the myths.

Reviewer #2: The article addresses an important matter regarding participation in screening mammography programs. Participation in screening programs is mostly affected by knowledge/awareness, access and personal beliefs. Being a smoker or alcohol consumer, BMI etc can have a direct relation with increasing breast cancer risk, but PARTICIPATING or not participating in a free mammography screening program due to these factors doesn't make sense. Also p-values and data could not be found so exact significance of these variables could not be ascertained.

6. PLOS authors have the option to publish the peer review history of their article (what does this mean?). If published, this will include your full peer review and any attached files.

Reviewer #1: No

Reviewer #2: No

---

## [Author Response · Author response to Decision Letter 0]

30 May 2022

Response to the editor:

Dear Mr. Soofi,

Thank you for considering our manuscript entitled “Determinants of mammography screening participation – A cross-sectional analysis of the German population-based Gutenberg Health Study (GHS)” by Pokora, Büttner, Schulz, Schuster, Merzenich, Teifke, Michal, Lackner, Münzel, Zeissig, Wild, Singer, and Wollschläger for publication and thank you for your good and valuable comments. The comments to the reviewer are in the document ‚Response to Reviewers‘. We responded all the comments and we included our comments to you in this cover letter. The comments to you are in black font, our replies in black font italic and underlined:

https://journals.plos.org/plosone/s/fileid=ba62/PLOSOne_formatting_sample_title_authors_affiliations.pdf".

Answer: Thank you. We revised the file naming.

“The GHS is funded through contract AZ 961-386261/733 from the government of Rhineland-Palatinate (“Stiftung Rheinland-Pfalz für Innovation”); the research programs “Wissen schafft Zukunft” and “Center for Translational Vascular Biology” of the Johannes Gutenberg University Mainz; and its contract with Boehringer Ingelheim and Philips Medical Systems, including an unrestricted grant for the GHS. This study was also supported by grant BMBF 01EO1503 from the Federal Ministry of Education and Research.”

Answer: We included the role the funders took in the study and added the sentence.

Change of financial disclosure statement:

“The GHS is funded through contract AZ 961-386261/733 from the government of Rhineland-Palatinate (“Stiftung Rheinland-Pfalz für Innovation”); the research programs “Wissen schafft Zukunft” and “Center for Translational Vascular Biology” of the Johannes Gutenberg University Mainz; and its contract with Boehringer Ingelheim and Philips Medical Systems, including an unrestricted grant for the GHS. This study was also supported by grant BMBF 01EO1503 from the Federal Ministry of Education and Research. The funders had no role in study design, data collection and analysis, decision to publish, or preparation of the manuscript."

“Dr. Wild reports grants and personal fees from Boehringer Ingelheim, grants and personal fees from Novartis Pharma, grants from Philips Medical Systems, grants from Bayer AG, grants and personal fees from Sanofi-Aventis, grants and personal fees from Bayer Vital, grants and personal fees from Daiichy Sankyo, grants and personal fees from Bayer Health Care, personal fees from AstraZeneca, personal fees and non-financial support from DiaSorin, non-financial support from I.E.M., outside the submitted work; Dr. Schuster reports the professorship for ophthalmic healthcare research endowed by ‘Stiftung Auge’ and financed by ‘Deutsche Ophthalmologische Gesellschaft’ and ‘Berufsverband der Augenarzte Deutschlands e.V.’ Schuster AK received research funding from Allergan, Bayer Vital, Novartis, PlusOptix and Heidelberg Engineering; Dr. Singer reports personal fees from Pfizer, personal fees from Bristol-Myers Squibb, personal fees from Boehringer-Ingelheim, personal fees from Lilly, outside the submitted work; Dr. Wollschläger reports grants from German Federal Ministry of Education and Research, during the conduct of the study. All other authors declare no conflict of interest.”

 Answer: We included the role the funders took in the study and added the sentence.

Change of competing interest statement: 

“Dr. Wild reports grants and personal fees from Boehringer Ingelheim, grants and personal fees from Novartis Pharma, grants from Philips Medical Systems, grants from Bayer AG, grants and personal fees from Sanofi-Aventis, grants and personal fees from Bayer Vital, grants and personal fees from Daiichy Sankyo, grants and personal fees from Bayer Health Care, personal fees from AstraZeneca, personal fees and non-financial support from DiaSorin, non-financial support from I.E.M., outside the submitted work; Dr. Schuster reports the professorship for ophthalmic healthcare research endowed by ‘Stiftung Auge’ and financed by ‘Deutsche Ophthalmologische Gesellschaft’ and ‘Berufsverband der Augenarzte Deutschlands e.V.’ Schuster AK received research funding from Allergan, Bayer Vital, Novartis, PlusOptix and Heidelberg Engineering; Dr. Singer reports personal fees from Pfizer, personal fees from Bristol-Myers Squibb, personal fees from Boehringer-Ingelheim, personal fees from Lilly, outside the submitted work; Dr. Wollschläger reports grants from German Federal Ministry of Education and Research, during the conduct of the study. All other authors declare no conflict of interest. This does not alter our adherence to PLOS ONE policies on sharing data and materials.”

Answer: Unfortunately, we cannot provide a public minimum data set for data protection reasons. In principle, the GHS is subject to the regulations of the general data protection regulation (DSGVO) when passing on personal data. The data may not be forwarded to third parties without the consent of the participants. At the GHS, there is a separate consent for disclosure to cooperation partners. However, if this consent is given, the data can only be passed on in the context of a scientific cooperation if the GHS Steering Committee deems it useful and necessary from a scientific point of view. However, there are also participants who have not agreed in principle to data being passed on. For them, it does not go under any circumstances. For this reason, we have updated our Data Availability statement as follows.

"The GHS data cannot be made publicly available because participants were assured that the data would only be used for scientific research purposes, and therefore the informed consent included only limited data sharing. At GHS, there is a separate consent to share data with collaborators. However, if this consent is given, the data can only be shared as part of a scientific collaboration if the GHS Steering Committee deems it appropriate and necessary from a scientific perspective. However, there are also participants who have not consented to the sharing of data in principle. For these participants, data can only be analyzed on-site. Available data are available to researchers who meet the criteria for access to confidential data from the GHS Coordinating Principal Investigator (philipp.wild@unimedizin-mainz.de). More detailed contact information can be found on the home pages of the GHS

(http://www.gutenberghealthstudy.org/ghs/overview.html?L=1)."

Thank you. We reviewed the reference list and corrected the mistakes.

Best regards,

Roman Pokora

Response to the reviewer:

Thank you for reviewing our manuscript. Please find attached our point-by-point response to the comments. Reviewer comments are in black font, Author replies in black font italic and underlined.

1. Is the manuscript technically sound, and do the data support the conclusions?

Reviewer #1: Yes

Reviewer #2: Yes

2. Has the statistical analysis been performed appropriately and rigorously?

Reviewer #1: I Don't Know

Reviewer #2: I Don't Know

3. Have the authors made all data underlying the findings in their manuscript fully available?

Reviewer #1: Yes

Reviewer #2: No

4. Is the manuscript presented in an intelligible fashion and written in standard English?

Reviewer #1: Yes

Reviewer #2: Yes

5. Review Comments to the Author

Reviewer #1: It is a technically sound study, well conducted and a well written manuscript. Would be nice to add a little detail about the mammography invitation process for the non-German reader. 

Answer: We added in the introduction (line 67ff) a part about the invitation process.

“Eligible women are informed every two years by an invitation letter from one of 14 Central Offices which organize the mammography screening program nationwide. A detailed information brochure is sent out with the invitation. The brochure serves as a decision-making aid and provides information on the examination procedure, on breast cancer, and on possible advantages as well as disadvantages of participating in a screening program.”

I feel it would be important to explore perhaps as a separate study, the reasons behind the decrease in mammography (recently) with higher education. Are they accessing bad information on social media or misinterpreting information about radiation exposure etc. This may inform further decisions and the invitation process may need to have more detailed information on dispelling the myths.

Answer: We agree with the reviewer that specifically the groups that do not participate would need to be examined more closely in a qualitative study. It would also be interesting to find out where these groups get their information.

Reviewer #2: The article addresses an important matter regarding participation in screening mammography programs. Participation in screening programs is mostly affected by knowledge/awareness, access and personal beliefs. Being a smoker or alcohol consumer, BMI etc can have a direct relation with increasing breast cancer risk, but PARTICIPATING or not participating in a free mammography screening program due to these factors doesn't make sense. 

Answer: In the absence of better variables, we included the variables smoking, BMI, alcohol consumption, etc. as a proxy for health behaviors in our model.

Also p-values and data could not be found so exact significance of these variables could not be ascertained.

Answer: We understand the concerns from the reviewer. We do not report p-values anywhere in the manuscript. The reasons for doing so:

1. The sample size is sufficiently large that even seemingly trivial differences are likely to be statistically significant. 

2. P-values are commonly misinterpreted as the probability that the test hypothesis is true, or as the probability that observed association is due to chance alone. Both are false.

In sum, in the present study we do not believe that adding P values would help readers to get a better sense of whether the reported characteristics differ across women, or that p-values and significance testing would add valuable information beyond the reported confidence intervals.

We hope that the explanations will satisfy the reviewers and that the editors will appreciate our efforts to conform with the STOBE statement and the position of the American Statistical Association (2016). Enclosed we send the result tables with the p-values.

Thank you for reviewing our manuscript and for the helpful comments. We are looking forward to your reply.

best regards on behalf of all authors

Roman Pokora

---

## [Decision Letter · Decision Letter 1]

19 Sep 2022

Determinants of mammography screening participation – A cross-sectional analysis of the German population-based Gutenberg Health Study (GHS)

PONE-D-21-26679R1

Dear Dr. Pokora,

We’re pleased to inform you that your manuscript has been judged scientifically suitable for publication and will be formally accepted for publication once it meets all outstanding technical requirements.

Kind regards,

Edward Jay Trapido, ScD

Academic Editor

PLOS ONE

Additional Editor Comments (optional):

Reviewers' comments:

Reviewer's Responses to Questions

**Comments to the Author**

1. If the authors have adequately addressed your comments raised in a previous round of review and you feel that this manuscript is now acceptable for publication, you may indicate that here to bypass the “Comments to the Author” section, enter your conflict of interest statement in the “Confidential to Editor” section, and submit your "Accept" recommendation.

Reviewer #2: All comments have been addressed

Reviewer #3: (No Response)

2. Is the manuscript technically sound, and do the data support the conclusions?

Reviewer #2: Yes

Reviewer #3: Yes

3. Has the statistical analysis been performed appropriately and rigorously? 

Reviewer #2: I Don't Know

Reviewer #3: No

4. Have the authors made all data underlying the findings in their manuscript fully available?

Reviewer #2: Yes

Reviewer #3: No

5. Is the manuscript presented in an intelligible fashion and written in standard English?

Reviewer #2: Yes

Reviewer #3: Yes

6. Review Comments to the Author

Reviewer #2: (No Response)

Reviewer #3: (No Response)

7. PLOS authors have the option to publish the peer review history of their article (what does this mean?). If published, this will include your full peer review and any attached files.

Reviewer #2: No

Reviewer #3: No

---

## [Editor Report · Acceptance letter]

26 Sep 2022

PONE-D-21-26679R1 

Determinants of mammography screening participation – A cross-sectional analysis of the German population-based Gutenberg Health Study (GHS) 

Dear Dr. Pokora:

I'm pleased to inform you that your manuscript has been deemed suitable for publication in PLOS ONE. Congratulations! Your manuscript is now with our production department. 

Kind regards, 

on behalf of

Dr. Edward Jay Trapido 

Academic Editor

PLOS ONE